# Shifting of the Migration Route of White-Naped Crane (*Antigone vipio*) Due to Wetland Loss in China

Yifei Jia [1,2], Yunzhu Liu [2], Shengwu Jiao [3], Jia Guo [1,2], Cai Lu [1,2], Yan Zhou [4], Yuyu Wang [1,2], Guangchun Lei [1,2,*], Li Wen [5,6] and Xunqiang Mo [7]

1   Centre for East Asian-Australasian Flyway Studies, Beijing Forestry University, Beijing 100083, China; jiayifei@bjfu.edu.cn (Y.J.); guojia.eco@foxmail.com (J.G.); lucai.wetland@foxmail.com (C.L.); wangyy@bjfu.edu.cn (Y.W.)
2   School of Ecology and Nature Conservation, Beijing Forestry University, Beijing 100083, China; rosie312@163.com
3   Research Institute of Subtropical Forestry, Wetland Ecosystem Research Station of Hangzhou Bay, Chinese Academy of Forestry, Hangzhou 311400, China; qustjsw@163.com
4   Co-Innovation Center for Sustainable Forestry in Southern China, College of Biology and the Environment, Nanjing Forestry University, Nanjing 210037, China; zhouyan.eco@foxmail.com
5   NSW Department of Planning, Industry and Environment, Science, Economics and Insights Division, Sydney 2150, Australia; li.wen@environment.nsw.gov.au
6   Department of Earth and Environmental Sciences, Macquarie University, Sydney 2109, Australia
7   School of Geographic and Environmental Sciences, Tianjin Normal University, Tianjin 300387, China; 421973@163.com
*   Correspondence: guangchun.lei@foxmail.com; Tel.: +86-010-62336717

**Abstract:** In the last 15 years, the west population of white-naped crane (*Antigone vipio*) decreased dramatically despite the enhanced conservation actions in both breeding and wintering areas. Recent studies highlighted the importance of protecting the integrity of movement connectivity for migratory birds. Widespread and rapid landcover changes may exceed the adaptive capacity of migrants, leading to the collapse of migratory networks. In this study, using satellite tracking data, we modeled and characterized the migration routes of the white-naped crane at three spatial levels (core area, migratory corridor, and migratory path) based on the utilization distribution for two eras (1990s and 2010s) spanning 20 years. Our analysis demonstrated that the white-naped crane shifted its migratory route, which is supported by other lines of evidences. The widespread loss of wetlands, especially within the stopover sites, might have caused this behavioral adaptation. Moreover, our analysis indicated that the long-term sustainability of the new route is untested and likely to be questionable. Therefore, directing conservation effects to the new route might be insufficient for the long-term wellbeing of this threatened crane and large-scale wetland restorations in Bohai Bay, a critical stopover site in the East Asian-Australasian flyway, are of the utmost importance to the conservation of this species.

**Keywords:** landcover change; migratory behavior flexibility; satellite tracking; stopover site; utilization distribution; white-naped crane

## 1. Introduction

Rapid population and economic growth across the globe, and the associated increase in demands on natural resources, particularly land and water, have put tremendous pressures on the habitats required by migratory birds [1]. As a result, the population of migratory birds is continuously declining [2–5]. This disturbing global pattern [6] has raised concerns of an impending wave of extinctions [7,8]. Recent studies continue to highlight that sustaining the integrity of migratory connectivity is fundamental for successfully conserving migratory species, and that management actions focused on breeding and wintering area alone are not sufficient to curb the population decline of threatened species [4,9–11].

Migratory waterbirds, such as geese and swans, often have high site fidelity including breeding, staging, and wintering sites [12–15] and many birds return to the same sites year after year. This site faithfulness is undoubtedly important for the stability of the migratory network [4]. However, many of the current stopover sites are threatened due to increased urbanization, agriculture, and others anthropogenic activities [2,16], and the widespread loss of stopover habitat is recognized as a key contributing factor in the population decline of many migratory bird species [1,17,18]. Thus, the ability to respond to environmental changes at stopover sites may also be an important aspect of migration, and species with restricted habitat choice or few available stopover areas could suffer more than species with flexible food choice and a selection of many stopover sites [19].

Migration is universal in waterbirds and they exhibit diverse migratory strategies [20]. Although waterbirds have some degree of behavioral flexibility, and can adapt to environmental changes by, for example, adjusting their diet, selecting different breeding and wintering sites, and the timing of migration onset [14,21,22], large-scale (e.g., flyway) and rapid landcover alteration may exceed the adaptive capacity of many migrants and cause the collapse of migratory networks [4]. For example, in the East-Asia Australasia flyway (hereafter EAAF), the dramatic loss of wetlands in China's Northeast Plains [23] has been identified as a critical migration impediment for geese [11,24] and the habitat degradation in Bohai Bay region was the main cause of population decline in several shorebirds [2]. As a result, the EAAF is currently the most threatened flyway in the world, and many of its species (19%) are categorized as endangered on the IUCN Red List [1].

Cranes are impressive migrants and excellent navigators [25]. Due to their large body size, cranes are among the first birds to be utilized in satellite-tracking studies [26]. Harris [27] tracked six white-naped cranes (hereafter referred to as the crane, *Antigone vipio*) in the early 1990s using satellite transmitters to map and describe the migratory paths and the main stopover locations. In the early 2010s, the International Crane Foundation (ICF), Mongolian Academy of Sciences (MAS) and Wildlife Science and Conservation Center of Mongolia (WSCC) conducted another campaign of tracking. In the 1990s, China's economy began to boom. The rapid economic growth and the associated landcover change peaked in the mid 2010s. Therefore, the two datasets provide a unique opportunity to investigate how the crane responded to rapid environmental change and, more generally, to study the migratory behavior flexibility of the crane [22].

With fewer than 6000 individuals [28], the crane has been listed as vulnerable on the IUCN Red List since 1994 [29]. The western population, which winters in the floodplain lakes of the middle-lower Yangtze region, has decreased sharply from 4000 individuals in 2002 to 1000–1500 individuals in the past 15 years [28] despite the strengthening of protection policy in both breeding and wintering areas. The comparison of the migration features of the two phases could provide insights on the causes of the observed rapid population decline and enable management to identify pressures and to prioritize actions along the entire flyway. In these contexts, the objectives of the study are to: (i) model and characterize migration routes of the crane in the 1990s and 2010s with the same analysis framework; (ii) investigate the extent of changes in the cranes' migratory behaviors, including shifts in stopover site fidelity, migratory paths, and duration of stopover; and (iii) identify the causes of migratory behavior changes by mapping and comparing the landcover compositions within the two modeled migratory routes. The working hypothesis of this study is that if there are large transformations in landcover types at the pathway scale, the crane would show behavioral adaptation to these environmental changes. Based on our results, we also discuss the responses to future changes and the management options for efficiently conserving this population.

## 2. Data and Methods

### 2.1. White-Naped Crane Population Census

The white-naped crane breed in the Daurian Steppe, Amur and Ussuri on the border of Russia, Mongolia and China, with the western population wintering in China and the

majority of the eastern population wintering in Korea and Japan [30]. All the western population, and a small fraction of the eastern population, use the wintering grounds in the Yangtze River Basin, mainly at Poyang Lake. The continuous decline of the western population became a cause of concern over the past two decades [31].

During the migratory seasons in 2012–2016, we conducted a census of the western crane population at known stopover sites (i.e., Duolun County, Miyun Reservoir, Beidahang Reservoir, Beidaihe Reserve, Huanzidong Reserve, and Yellow River Delta Reserve) and wintering grounds (Poyang Lake). We employed a point count method for all sites using a spotting scope (Swarovski ATS 80 HD 20-60). We counted birds at a fixed high vantage point and attempted to achieve a full view of the habitats from the observation point. In large wetlands, more than one observation point was selected for counting to ensure that the entire habitat was visible and surveyed.

### 2.2. GPS Data Gathered in 1990s

From 1991–1993, six white-naped cranes were captured in Daursky Nature Reserve (NR) (50°N, 115°E) and the Mangut region (50°N, 112°E) [27,32]. The captured cranes were mounted with satellite transmitters (Platform transmitter terminals, PTT) model T-2050 (Nippon Telegraph and Telephone Corporation, Tokyo, Japan and Toyo Communication Equipment Corporation, Tokyo, Japan, weight: 43 g). The PTTs were harnessed to the backs of the cranes with Teflon-treated ribbons as described in [26]. The transmitters were set to 6 h active and 12 h inactive with 60 s between pulses [33]. We used the telemetric data for six cranes which migrated to Poyang Lake for wintering (Table 1).

**Table 1.** Information on the tracked white-naped cranes.

| Year | ID | Age | Type of Tracker | Migration Start Date [1] | Migration End Date [2] | Days | Number of Fixes | Accuracy (m) |
|---|---|---|---|---|---|---|---|---|
| 2013 | X06 | Adult | GPS-GSM | 1 Sep. 2013 | 30 Nov. 2013 | 90 | 3555 | <125 |
| 2014 | X06 | Adult | GPS-GSM | 1 Sep. 2014 | 2 Nov. 2014 | 62 | 1145 | <125 |
| 2014 | T2 | Juvenile | GPS-GSM | 10 Oct. 2014 | 3 Nov. 2014 | 23 | 1070 | <125 |
| 2014 | T3 | Juvenile | GPS-GSM | 22 Sep. 2014 | 2 Nov. 2014 | 40 | 1976 | <125 |
| 2014 | T4 | Adult | GPS-GSM | 15 Sep. 2014 | 5 Dec. 2014 | 80 | 3207 | <125 |
| 2014 | T5 | Adult | GPS-Argos | 4 Oct. 2014 | 2 Nov. 2014 | 29 | 229 | <100 |
| 2014 | T6 | Adult | GPS-Argos | 6 Oct. 2014 | 2 Nov. 2014 | 27 | 207 | <100 |
| 1992 | 9375 | Adult | GPS-Argos | 8 Oct. 1992 | 1 Nov. 1992 | 22 | 30 | 350–1000 |
| 1991 [3] | 9377 | Unknown | GPS-Argos | 20 Oct. 1991 | 5 Nov. 1991 | 26 | 7 | 350–1000 |
| 1993 | 20248 | Adult | GPS-Argos | 9 Oct. 1993 | 30 Oct. 1993 | 21 | 33 | 350–1000 |
| 1993 | 20250 | Adult | GPS-Argos | 11 Oct. 1993 | 14 Nov. 1993 | 33 | 55 | 350–1000 |
| 1993 | 20252 | Adult | GPS-Argos | 17 Oct. 1993 | 16 Dec. 1993 | 64 | 99 | 350–1000 |
| 1993 | 20253 | Adult | GPS-Argos | 16 Oct. 1993 | 22 Nov. 1993 | 36 | 51 | 350–1000 |

[1] "Migration start date" is the date when the crane left the breeding ground; [2] "Migration end date" means the date when the crane arrived at the wintering ground. Usually, cranes fly several hundred kilometers during migration; [3] Excluded due to too few useful relocations.

### 2.3. GPS Data Collected in 2013–2014

The Khurkh and Khuiten Valleys (48°22′28.2″N 110°21′32.5″E, 48°17′15.9″N 110°5′17.9″E) in Mongolia are two major nesting sites for the crane. Between August 2013 and 2014, ICF, MAS, and WSCC carried out tracking work. A total of seven cranes were tracked: two were mounted with GPS-Argos tags (PTTs, North Star Science and Technologies, King George, VA, USA; model: 30 GPS; weight: 30 g) and five were equipped with GPS-GSM tags (CTTs, Cellular Tracking Technologies Rio Grande, NJ, USA; model: CTT-1060a; weight: 23 g). All tracking devices were fixed during the molting period when the birds are unable

to fly. The Argos-GPS system was set to record the location once every 8 h. The GPS-GSM systems recorded the position of the crane once every 30 min. The PTT data were downloaded daily via the Argos Message Retriever (PTT Tracker Ver1.0.0.3.7, GeoTrak, Inc. www.geotrakinc.com/ptt-tracker/ (accessed on 13 January 2018)). In order to provide higher location accuracy and increase the number of available positions, the tags were programmed to record Argos and GPS location simultaneously [34]. Each Argos record was assigned to one of the seven location classes (LC): 3, 2, 1, 0, A, B, or Z. Each LC was associated with an estimated error based on the number of messages received per satellite pass: from <150 m to >1000 m (LC 3, 2, 1, and 0), no accuracy estimation (A and B), and invalid location (Z). Records with high errors (i.e., LC 0, A, B, and Z) were excluded from modelling. The CTT data were downloaded from the account.celltracktech.com/ (accessed on 13 January 2018). The GPS location accuracy estimation (HDOP, fix $\geq$ 3) was from <25 m to 76–100 m.

### 2.4. Telemetric Data Processing

We first removed duplicated records from the raw telemetry data. We then created 12 trajectories with the cleaned and quality assured records (one for each tagged bird: five from the 1991–1993 database and seven from the 2013–2014 database; Table 1). The R package adehabitatLT [35] was used to create, manipulate, and store the trajectories. Although the PTT and CTT were coded to return regular relocations, minor delays often occurred, resulting in an irregular sequence. In addition, missing location points were common when the GPS unit failed to receive signals [36]. More importantly, the sampling intervals of the two satellite tracking campaigns were very different (i.e., 6/12 h and 0.5–8 h for the 1991–1993 and 2013–2014, respectively). To ensure the modelled migratory paths for the two periods were comparable, the time step of all trajectories was resampled to 6 h and the missing points were filled by linear interpolation prior to further analysis.

### 2.5. Habitat Utilization Modelling

As temporal autocorrelation is common and an intrinsic property of animal relocation sequential data [37,38], we used the biased random bridge (BRB) approach to estimate the utilization distribution (UD) of the white-naped cranes. This approach is similar to the Brownian bridge approach [39], with several improvements [40]. The Brownian bridge estimates the density of probability that a trajectory passes through any point of the study area. The Brownian bridge is built on a conditional random walk between successive pairs of relocations, dependent on the time between locations, the distance between locations, and the Brownian motion variance that is related to the animal's mobility [40]. This assumes that the animal movement is random and purely diffusive between two successive relocations: it is supposed that the animal moves in a purely random fashion from the starting relocation and reaches the next relocation. The BRB approach goes further by adding an advection component (i.e., a "drift") to the purely diffusive movement in the Brownian bridge: it assumes that the animal movement is governed by a drift component (i.e., a general tendency to move in the direction of the next relocation) and a diffusion component (tendency to move in other directions than the direction of the drift) [41]. The addition of the drift is therefore considered more realistic in modelling animal movements [42]. The detailed description of BRB approach can be found in Benhamou [39] and was briefly outlined as follows:

Considering one step in an animal trajectory includes two successive relocations $r_1 = (x_1, y_1)$ and $r_2 = (x_2, y_2)$ collected at time $t_1$ and $t_2$, the BRB estimates the probability density function (PDF) of the animal located at any place $r_i = (x_i, y_i)$ at time $t_i$ (with $t_1 < t_i < t_2$). Benhamou (2011) noted that the BRB can be approximated by a bivariate normal distribution:

$$f(r_i, t_i | r_1, r_2) = \frac{t_2 - t_1}{4\pi D(t_2 - t_1)} exp\left[\frac{r_m D r_m}{4 p_i (t_2 - t_1)}\right] \quad (1)$$

$$r_m = \frac{x_1 + p_i(x_2 - x_1)}{y_1 + p_i(y_2 - y_1)} \tag{2}$$

$$p_i = \frac{t_i - t_1}{t_2 - t_1} \tag{3}$$

where $r_m$ is the mean location, $p_i$ is the proportion of time from starting relocation to $r_m$, and $D$ is the diffusion matrix.

From the BRB model, UD and core area were defined as the areas encompassed within 95% and 50% UD isopleths, respectively [43]. We also calculated the "flatness" of the UD, defined as the ratio core area/home range area for each tagged white-naped crane. These variables were subjected to further analysis.

We compared the UD modelled for each crane at the same period using the Schoener's D index [44], which is a classic and reliable measure of niche overlap (Rödder and Engler, 2011) widely used in ecological studies. The Schoener's D ranges from 0 (distribution models have no overlap) to 1 (distribution models are identical) and are derived from the difference in probability distributions over space produced between two ecological niches. The UD for the four cranes in 2014 was almost identical according to the criteria of [45] (Schoener's D ranged from 0.97 to 0.98 for all paired comparisons). Similarly, the UD of the five cranes in 1991–1993 also had a very high level of overlap (Schoener's D ranged from 0.91 to 0.95 for all paired comparisons). Based on these, we averaged the UDs in each period and produced two migratory paths (one for 1993 and another for 2014). We classified the average raster into three levels of UD using Jenks natural breaks optimization [46] in ESRI ArcGIS 10.1: core area, migratory corridor, and migratory path. We referred to the estimated home range (a term used for resident animals, e.g., [47]), which encompasses 95% of the modelled UD, as the migratory path for the period of migration. In this study, all areas with a medium level of UD within the migratory path were referred to as the migratory corridor. The core area, corresponding to the most intensively used areas within the estimated migratory paths, were referred to as the stopover site for the migrants to refuel [27,32].

### 2.6. Landcover Data

Landcover changes between 1990 and 2010 within the three classes of UD were used to understand the causes of changes in the main stopovers and the shift of migratory paths. The 30 m resolution Landsat TM satellite images from (landsat.usgs.gov/ (accessed on 26 February 2016)) and HJ-1A/B satellite images from (218.247.138.119:7777/DSSPlatform/index.html (accessed on 26 February 2016)) were used to map landcover in 1990 and 2010. Using the landcover classification system, we classified the landcover in China into 38 categories. We used an object-oriented automated algorithm, combined with ground survey and radar data, to classify the land use of 2010. Based on the 2010 landcover classification, the landcover in 1990 was produced using object-oriented vector similarity monitoring methods.

To facilitate comparison, the 38 landcover classes were grouped into six main classes: grassland, wetlands, open waters (lakes and reservoirs), croplands, developed area (residential, roads, mining, and industrial areas), and forests. We calculated and compared the suitable crane habitats within each class of utilization level for the two migratory paths. Three landcover types are assumed to be suitable habitats: wetlands, grasslands, and croplands [27,32]. We excluded open waters (include lakes and reservoirs) as they are too deep for the crane [27].

## 3. Results

### 3.1. White-Naped Crane Abundance at Different Sites along the Migratory Route

Table 2 presents the highest bird counts at sites along the western migratory routes for the period of 2012–2016 together with historical records. There was a sharp decline in the western population that winters in China (a reduction of 66.5% compared with historical record; Table 2). The data also showed the disappearance or shrinkage of some traditional

stopover sites in the Northern-east Plain (i.e., Dalinor Lake and Beidaihe Reserve, Figure 1) and the appearance (or discovery) of new stopover sites (e.g., Duolun County, Miyun Reservoir, Figure 1).

**Table 2.** The highest count of white-naped crane (2012–2016) in different sites (Figure 1).

| Sites | Survey Dates | Count | History Records Dates | Count |
|---|---|---|---|---|
| Beidaihe Reserve | Oct. 2013 | 0 | Oct. 1985 | 152 |
| Dalinor Lake | - | - | 1980s | >100 [1] |
| Yellow River Delta Reserve | Nov. 2013 | 65 | Nov. 2004 | 600 [2] |
| Beidagang Reservoir | Nov. 2013 | 117 | - | - |
| Duolun County | Oct. 2014 | 663 | - | - |
| Miyun Reservoir | Mar. 2013 | 1342 | - | - |
| Poyang Lake | Jan. 2012 | 738 | 1986 | 2200 [1] |

[1] Proceedings of the 1987 international crane workshop, www.savingcranes.org/wp-content/uploads/2008/05/Proceedings-of-the-1987-International-Crane-Workshop.pdf (accessed on 3 December 2017); [2] Yellow River delta survey data; - no data.

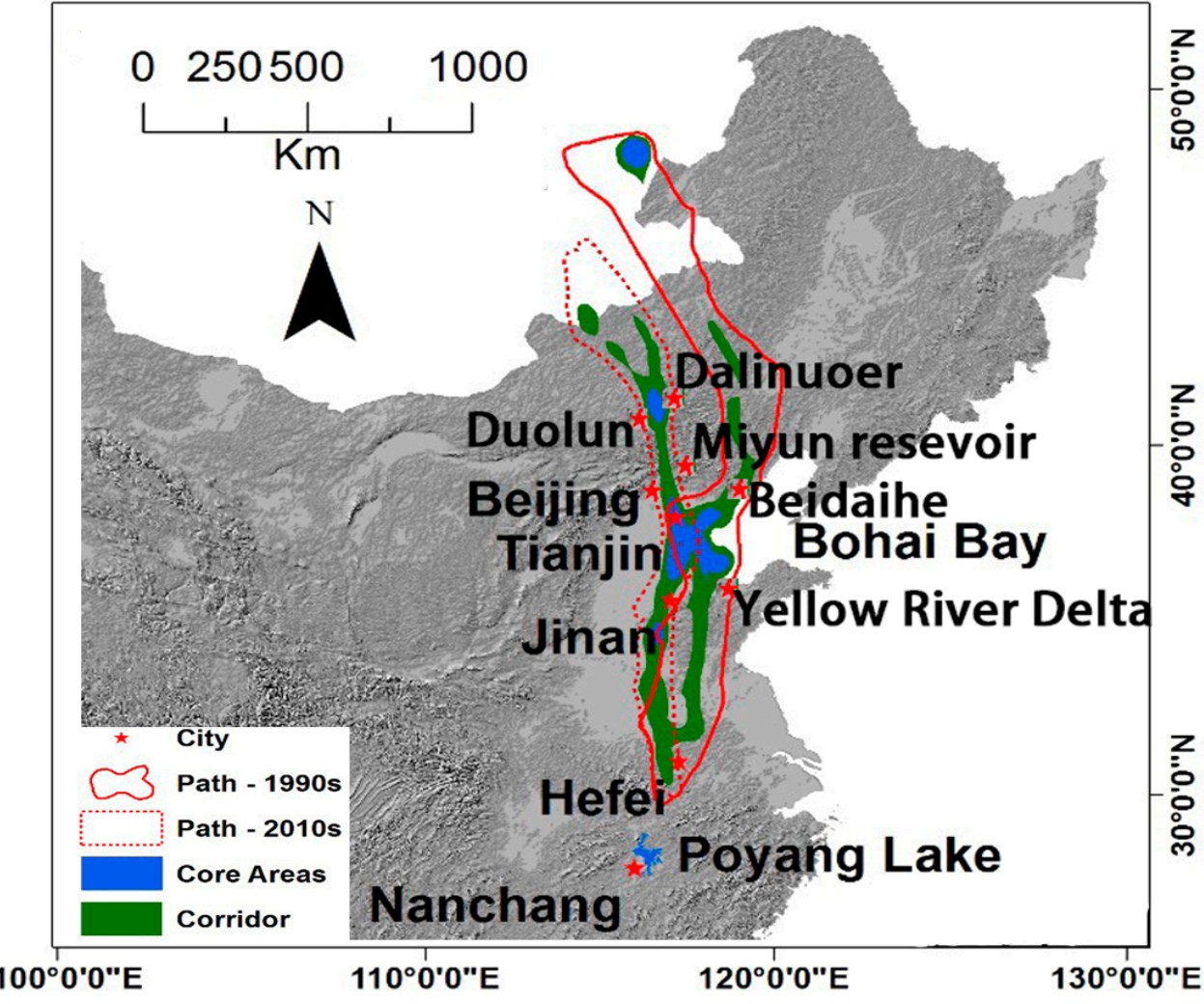

**Figure 1.** The extended migratory paths of the white-naped crane during the early 1990s (solid red) and early 2010s. The paths and core areas are in green and blue, respectively. The 1 km hill shade shows the boundary of China.

### 3.2. Migratory Paths at the Early 1990s and 2010s

Figure 1 presents the modelled migratory paths of white-naped cranes for the two periods. Two independence paths were evident for all three levels of utilization distribution. In particular, there was no overlap between the migratory paths in the two periods further north of Bohai Bay, after which the two routes started to converge, and became entirely overlapped at the vicinity of Hefei as the cranes headed to their wintering site at Poyang Lake (Figure 1). For simplicity, we will refer to the two passages as the Beidaihe route (1990s) and the Duolun route (2010s).

The size of modelled UD was larger in the 1990s than in the 2010s for all three levels of utilization. The stopover sites especially, defined as the most intensive used area, were reduced by 15.70% (Table 3). Moreover, the flatness of the migratory path was increased by 13.99% (Table 2), indicating an increasingly explorative behavior in 2010s compared with in 1990s.

**Table 3.** Summary the migratory paths of white-naped cranes in the early 1990s and early 2010s.

|  | Beidaihe (1990s) | Duolun (2010s) | Change (%) |
|---|---|---|---|
| Stopover (ha) | 2,029,093 | 1,710,596 | 15.7 |
| Corridor (ha) | 10,430,587 | 9646,780 | 7.51 |
| Path (ha) | 29,899,740 | 22,093,500 | 26.11 |
| Flatness | 6.79 | 7.74 | −13.99 |

Note: the marine area within the paths was excluded.

The Beidaihe path included three stopover sites, one at Ozero Tsagaan-Nuur in Russia and two in China (Dalinuor Lake and Bohai Bay, Figure 1). The Duolun path also included three stopover sites (Duolun, Bohai Bay, and Jinan, Figure 1). The areas in the vicinity of Tianjian, near Bohai Bay, were classified as stopover site for the white-naped cranes within the migratory paths for both periods (Figures 1 and 2). However, the commonly shared area was relatively small (17.4% and 17.1% of the stopover sites for 1990s and 2010s, respectively; Figure 2). The most noticeable changes between the two periods were that two areas at Bohai Bay that were identified as important stopover sites before were not included in the 2010s migratory path: Northern Bohai and the Yellow River Delta [27]. Our results showed the average time that the migratory crane spent in the two flyways in Bohai Bay was significantly different. Based on the arrival and departure date, the cranes stayed for $20 \pm 18.3$ days ($n = 6$) on the Beidaihe route, but only for $3 \pm 3.9$ days ($n = 7$) on the Duolun route.

### 3.3. Landcover Changes within the Beidaihe Route

Within the early migratory path in the 1990s, a total of 23,733,865 ha (79%) was classified as suitable crane habitat in 1990. This figure decreased to 22,766,534 ha (76%) in 2010. While the reduction in grassland was marginal, large areas of cropland and wetland were lost (a reduction of 160,160 ha or 29.84% and 954,457 or 7% for cropland and wetland, respectively), and the largest gain was in developed areas (with an increase of 1008,741 ha or 53%, Table 4).

The size of suitable habitat within the migratory corridor decreased by 376,544 ha (6%). There were reductions for all three broadly defined habitat types (2%, 11%, and 34% for grassland, cropland, and wetland, respectively; Table 4). Similar to the migratory path, there were large gains for developed areas (411,412 ha or 60%).

The most dramatic landcover changes were realized within the stopover sites. As we had no data for the stop over site at Ozero Tsagaan-Nuur in Russia, the values in Table 4 were for Bohai Bay only. From 1990 to 2010, a total of 148,155 ha (13%) of suitable habitat was lost. The loss of grassland, wetland, and cropland were 3038 ha (25%), 35,103 ha (42%), and 128,795 ha (20%), respectively. In the meantime, the developed area expanded from 102,731 ha to 224,343 ha (an increase of 118%; Table 4).

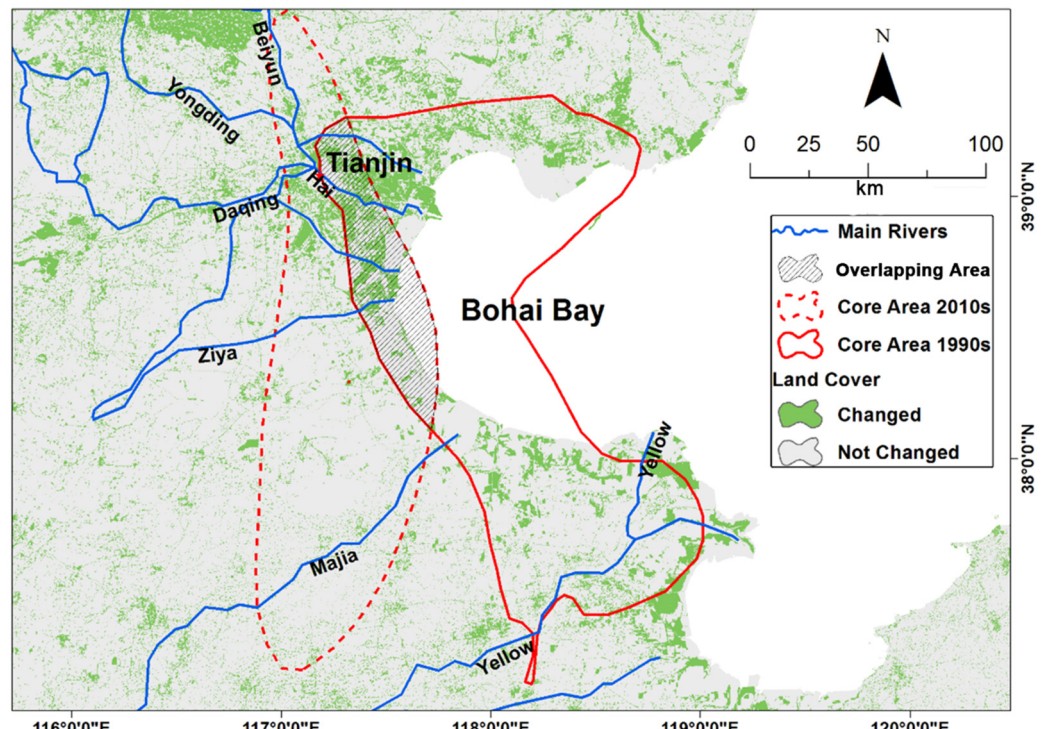

**Figure 2.** Map shows the small overlap (grey shaded area) of stopover sites between the migratory paths during the early 2010s (dashed red line) and early 1990s (solid red line) in the vicinity of Tianjin, the largest city on the shore of Bohai Bay. The background is the landcover changes (green) between 1990 and 2010.

**Table 4.** Landcover (ha) within the early Beidaihe migratory path.

| | Path | | | Corridor | | | Stopover Site [1] | | |
|---|---|---|---|---|---|---|---|---|---|
| | **1990** | **2010** | **± %** | **1990** | **2010** | **± %** | **1990** | **2010** | **± %** |
| Grasslands [2] | 7,739,334 | 7,690,019 | −1 | 760,817 | 747,610 | −2 | 12,338 | 9300 | −25 |
| Wetlands | 469,380 | 309,220 | −34 | 173,326 | 98,681 | −43 | 83,650 | 48,547 | −42 |
| Open water [3] | 1,426,891 | 1,623,492 | 14 | 597,056 | 755,500 | 27 | 378,156 | 413,035 | 9 |
| Cropland | 14,098,260 | 13,143,803 | −7 | 4,566,698 | 4,119,561 | −10 | 649,615 | 520,820 | −20 |
| Developed area | 1,900,592 | 2,909,333 | 53 | 680,438 | 1,091,850 | 60 | 102,731 | 224,343 | 118 |
| Bare ground | 552,892 | 497,554 | −10 | 122,102 | 115,447 | −5 | 44,092 | 42,897 | −3 |
| Forest | 3,712,392 | 3,786,617 | 2 | 1,010,777 | 1,038,067 | 3 | 19,448 | 29,171 | 50 |
| Total Habitat | 23,733,865 | 22,766,534 | −4 | 6,097,897 | 5,721,353 | −6 | 1,123,759 | 991,703 | −12 |
| Habitat (%) | 79 | 76 | −4 | 77 | 72 | −6 | 87 | 77 | −12 |

[1] Bohai site only, no data available for the site at Ozero Tsagaan-Nuur in Russia; [2] man-made lawns in urban areas are excluded; [3] marine and estuary areas are excluded.

### 3.4. Landcover Changes within the Duolun Migratory Path

The landcover changes within the current migratory path had similar patterns to those within the early path. Suitable habitats were reduced 6%, 7%, and 8% within the migratory path, corridor, and stopover sties, respectively (Table 5). There was a 38% reduction of wetland within the more extensive migratory path (34% and 21% for corridor and stopover sites, respectively; Table 5). The area of croplands also decreased considerably within all three levels of the white-naped crane utilization distribution, although the loss was not as dramatic as seen in the wetlands (Table 5). As with the path in the early 1990s, there were large increases in developed areas (50%, 49%, and 68% within the migratory path, corridor, and core areas, respectively, Table 5).

**Table 5.** Landcover (ha) within the current Duolun migratory path.

| | Path | | | Corridor | | | Stopover Site | | |
|---|---|---|---|---|---|---|---|---|---|
| | **1990** | **2010** | **± %** | **1990** | **2010** | **± %** | **1990** | **2010** | **± %** |
| Grasslands [1] | 5,817,544 | 5,762,763 | −1 | 2,002,656 | 1,972,565 | −2 | 269,150 | 264,477 | −2 |
| Wetlands | 241,842 | 149,240 | −38 | 143,618 | 95,317 | −34 | 59,461 | 49,069 | −17 |
| Open water [2] | 559,258 | 672,197 | 20 | 352,726 | 416,558 | 18 | 137,986 | 150,095 | 9 |
| Cropland | 7,873,217 | 7,058,420 | −10 | 4,816,241 | 4,309,400 | −11 | 1,134,427 | 1,008,662 | −11 |
| Developed area | 1,334,287 | 2,002,262 | 50 | 788,626 | 1,178,942 | 49 | 177,934 | 298,303 | 68 |
| Bare ground | 398,317 | 414,681 | 4 | 228,867 | 242,431 | 6 | 73,849 | 76,935 | 4 |
| Forest | 1,960,865 | 2,112,819 | 8 | 896,720 | 1,006,983 | 12 | 103,442 | 108,564 | 5 |
| Total Habitat | 14,491,861 | 13,642,620 | −6 | 7,315,241 | 6,793,841 | −7 | 1,601,024 | 1,472,303 | −8 |
| Habitat (%) | 80 | 75 | −6 | 79 | 74 | −7 | 82 | 75 | −8 |

[1] man-made lawns in urban areas are excluded; [2] marine and estuary areas are excluded.

### 3.5. Landcover Change within the Bohai Bay Stopover Sites

The size of the total suitable habitat within the stopover site was comparable for the two periods (1,123,759 ha and 1,026,608 ha for the 1990s and 2010s stopover sites, respectively; Table 6), and the percentage of suitable habitat within the stopover site was also similar. Moreover, the patterns of landcover change for the study period were similar, i.e., there was a large increase in development areas and a dramatic decrease in the size of wetlands and grasslands. However, there were large differences in the composition of the habitats. For example, there was 83,650 ha of wetland within the earlier stopover site, and only 36,742 ha of wetland in the 2010s stopover site.

**Table 6.** Landcover changes within the 1990s Bohai Bay stopover sites.

| Habitat | 1990 | 2010 | (± %) |
|---|---|---|---|
| Grasslands [1] | 12,338 | 9300 | −25 |
| Wetlands | 83,650 | 48,547 | −42 |
| Open water [2] | 378,156 | 413,035 | 9 |
| Cropland | 649,615 | 520,820 | −20 |
| Developed area | 102,731 | 224,343 | 118 |
| Bare ground | 44,092 | 42,897 | −3 |
| Forest | 19,448 | 29,171 | 50 |
| Total Habitat | 1,123,759 | 991,703 | −12 |
| Habitat (%) | 87 | 77 | −12 |

[1] man-made lawns in urban areas are excluded; [2] marine and estuary areas are excluded.

### 3.6. Landcover Change within the Beidaihe Migratory Path North of Bohai Bay

As all evidences point to the loss (or at least shrinkage) of part of the early migratory route (Table 2 and Figure 1), we explored the changes in landcover within the early migratory path north of Bohai Bay (Table 7). Although the total loss of total habitat was marginal (i.e., 3%; Table 7), more than 95% of the wetland area was lost between 1990 and 2010 (Table 7).

**Table 7.** Landcover change within the early migratory corridor north of Bohai.

| Habitat | 1990 (ha) | 2010 (ha) | Change (%) |
|---|---|---|---|
| Grasslands [1] | 544,805 | 528,711 | −16,094 (−3) |
| Wetlands | 9600 | 435 | −9165 (−95) |
| Open water [2] | 37,778 | 38,590 | 812 (2) |
| Cropland | 1,027,548 | 1,006,438 | −21,110 (−2) |
| Developed area | 114,518 | 161,900 | 47,382 (41) |
| Bare ground | 24,030 | 18,194 | −5835 (−24) |
| Forest | 607,791 | 615,435 | 7644 (1) |
| Total Habitat | 1,619,731 | 1,574,174 | −45,557 (−3) |

[1] man-made lawns in urban areas are excluded; [2] marine and estuary areas are excluded.

## 4. Discussion

In this study, we provided multiple lines of evidences to support the shifts in stopover sites and the migratory route of the western population of white-naped cranes: (1) the comparison of the satellite tracking data obtained in the early 1990s and 2010s; (2) multi-year population surveys; and (3) global white-naped crane population estimates (Wetland International at wpe.wetlands.org/) (accessed on 13 January 2020). There are a number of recent studies, which documented the diverse responses of migratory birds to environmental changes [22,48]. Fewer studies have addressed the flexibility of stopover sites and migratory routes [49]. These shifts might be a behavioral adaptation [50] to widespread habitat loss within the previous migratory route. To test this hypothesis, we mapped landcover in 1990 and 2010 using 30 m resolution satellite imagery and investigated the landcover changes between 1990 and 2010 within the two migratory paths. Our analysis indicated that the dramatic loss of wetlands in key stopover sites could be the decisive driver for the abandonment of previous stopover sites. The use of an alternative stopover site (i.e., Duolun) led to the westward shift of the migratory path (Figure 1).

### 4.1. Overall Landcover Change Patterns and Their Impacts on White-Naped Crane Migration

We found that the landcover change patterns were similar for the two migratory paths at all three utilization density levels, with a large increase in developed areas coinciding with China's economic growth during the past two decades. Overall, there was a reduction in suitable habitats within both paths, with the loss of wetlands being the largest. The results of this large-scale analysis provided limited support to our hypothesis, i.e., landcover change drives the shifts in stopover sites and migratory path. More localized analysis of the landcover changes for the path north of Bohai Bay (i.e., China's Northeast Plain) revealed a dramatic loss of wetlands (a 95% reduction in total wetland area; Table 7). In contrast, the reduction of wetlands in the north part of the Duolun route during the same period was much less at 41%. These changes might play a decisive role in the westward shift of the migratory route (Figure 1), which might contribute to the sharp decline of the western population of the white-naped crane (Table 2) and other migratory birds utilizing wetlands in the Northeast Plains [5]. Lei et al. [11] also identified the critical conservation gaps in the Northeast Plains for migratory geese.

### 4.2. Habitat Quality in Bohai Bay Region Could Be the Bottleneck of White-Naped Crane Conservation

Our analysis showed that Bohai Bay, where the previous and current migratory routes converge (Figure 1), is one of the most important stopover sites of the western crane population in China. The breeding ground of the crane in the Daurian steppe is vast and scattered. During autumn migration, the cranes fly south from their summering sites and congregate at Bohai Bay as a gathering place. All the investigated cranes wintering in the Yangtze Basin refueled here. With the disappearance of the Beidaihe route and the desertion of the Beidaihe Reserve and Dalinor Lake (Table 2) as stopover sites, this site has become more critical: there are no large wetland areas between Bohai Bay and the breeding

range of the crane along its migratory route. However, our results suggested that it was losing its function as a stopover site: the average duration of stopover was dramatically reduced from 20 days in the early 1990s to only 3 days in the early 2010s. The reduction in stopover duration could be explained by the deteriorated habitat quality: large areas of wetlands and natural grasslands in this region were converted into developed area (Table 5). Bohai Bay is also key stopover site for many other migratory waterbirds species such as shorebirds [51], storks [52] and geese [11]. It was also identified as bottleneck for shorebird migration passage [2,53,54].

### 4.3. The Possible Abandonment of the Beidaihe Route and Emergence of Duolun Route

Using satellite tracking, Higuchi et al. [33] mapped two migratory paths of white-naped cranes: from their breeding grounds in Russia, the east population flies to Izumi, Japan via the Kearon Peninsula, and the west population migrates to Poyang Lake via Bohai Bay (i.e., the Beidaihe path in this study). In this study, using the same satellite tracking technique, we found that the birds flew over the Xilin Gol Prairie and the valley of the Upper Luanhe River (main stopover sites) in China, before stopping briefly (for an average of 3 days) in Bohai Bay, and finally arriving in their wintering ground at Poyang Lake. We offer two alternative speculations for this obvious disagreement.

First, the cranes abandoned the stopover sites in China's Northeast Plains due to the dramatic reduction of wetlands (see below), shifting their stopover to the new Duolun area. Although the reduction of potential habitats (i.e., grasslands, croplands, and wetlands) within the Beidaihe route was marginal, the loss of wetlands was dramatic (nearly half of wetland area in the entire migratory route disappeared, Table 3; and 95% of the wetland area was converted to other landcover types north of Bohai Bay, Table 7). Cranes forage broadly (grassland, cropland, and wetland) at the stopover sites [26,55], however, they mainly roost in natural wetlands of sufficient size at night [56,57]. Despite the high site fidelity displayed by cranes [15,58], the dramatic loss of wetland habitat (thus limited safe night roosting sites) suggests they cannot stay in this area for long periods of time, which eventually led to the abandonment of the stopover sites in the Bohai Bay region. The multi-year population census data support this explanation (Table 1). The global white-naped crane population estimates provided another line of evidence: as the west population (i.e., those that winter in China) decreased, the east population (i.e., those that winter in Japan and Korea) gradually increased (Wetland International wpe.wetlands.org/ (accessed on 13 January 2020)). This also suggests that the cranes which used to take the Beidaihe route to Poyang Lake could now be migrating to Korea or Japan for wintering.

It is worth mentioning that the new Duolun route is located in semi-arid region, where climatic conditions are highly unpredictable [13]. Moreover, the predicted climate change is likely to increase the inter-and intra-annual variations in semi-arid regions [59]. As animal migrations are relatively regular events, of which many features, such as the onset, temporal patterns, and seasonal energy stores, are endogenously programmed [60], resource synchrony and predictability is critical for successful completion of the migratory journey [61]. Therefore, this low predictability could impose a critical risk for the sustainability of the Duolun route. Furthermore, wetlands in this region are often small and highly variable [62], suggesting that the environmental carrying capacity might be limited. In contrast, the previous Beidaihe route is situated in the Northeast Plains and North China Plains, which has a temperate humid or sub-humid continental monsoon climate [63], and historically supported extensive wetlands and grasslands [64]. In this context, migration along the Duolun route may be a sub-optimal choice for the white-naped cranes, which was partly confirmed by the increased flatness of migratory path (Table 2), and this behavioral adaptation might have complications for its long-term fit.

Second, as the number of tagged cranes was relatively small (six and seven for the 1990s and 2010s, respectively), both studies could potentially under-estimate the actual migratory path, and the two paths delineated in this study may be part of much broader pathway. As there was no crane recorded in the historical stopover sites of the Beidaihe

route (e.g., Beidaihe Reserve) in the recent population surveys (Table 1), we considered the likelihood of this scenario as low. Nevertheless, more observation data are needed to fully refute (or prove) this speculation.

Both speculations indicated that the habitat degradation within stopover sites poses a significant risk of losing migratory connectivity [10], which could lead to further decline of the western population of white-naped crane.

*4.4. Conservation Implications*

Tracking the annual migratory cycle provides insights into the processes which affect the population dynamics of migrants [10], enabling managers with essential knowledge and information to identify threats such as movement impediments [65] and the loss of important habitat sites [11], and allowing for the implementation of actions along the entire movement paths [66]. In this study, by comparing the migration trajectory of the white-napped crane for two periods spanning two decades, our results suggested that the western crane population might utilize an alternative stopover site, having shifted the migratory path westward to the Duolun route. The shift of migration route is likely the result of behavioral adaptation to large-scale habitat loss within the previous migratory path. The migratory population along the Duolun passage, especially at the Luanhe River Basin, the key stopover site, should be closely monitored.

Our analysis revealed that most of the breeding and wintering sites of the crane are currently protected as nature reserves, but the majority of stopover sites are located outside of the boundaries of current protected areas. Directing future conservation efforts to crucial stopover sites, such as maintaining sizable wetlands, and establishing and expanding protection areas in Duolun and Bohai Bay, should be an important part of the conservation strategy. More importantly, our analysis indicates that the long-term sustainability of the Duolun route is untested and likely to be questionable, thus, large scale wetland restorations in Northeast Plains stopover sites of the Beidaihe route are necessary for the long-term population viability of this threatened crane species and other migratory waterbirds [4,11].

## 5. Conclusions

Using multiple lines of evidences, we demonstrated the shift of the stopover and migratory path of the west population of white-naped cranes. The new route, which we referred as the "Duolun route", began at the breeding grounds in the Daurian Steppe, travelling south to the Luanhe River basin in Duolun county, China, and then took an easterly detour to Bohai Bay, before ending at the wintering site of Poyang Lake, southeast China. We found that the prevailing landcover change within the previous route (the Beidaihe route), especially the dramatic reduction of wetland areas in the stopover sites in China's Bohai Bay, was the decisive driver of this migratory route shift. As the sustainability of the Duolun route has not been tested and is likely low due to environmental unpredictability and the small carrying capacity resulting from wetland area limitations, the west population of white-naped crane may decrease further. Furthermore, the reduction of natural wetlands and grasslands in one of the key stopover sites, Bohai Bay, would further aggravate this situation. Based on these analyses, we proposed that conservation actions targeting key sites along the entire migratory path are required to efficiently conserve this population.

**Author Contributions:** Conceptualization, Y.J. and G.L.; methodology, L.W.; validation, Y.J., G.L. and L.W.; investigation, Y.L., J.G., S.J., X.M. and Y.Z.; writing—original draft preparation, Y.J. and Y.L.; writing—review and editing, Y.J. and Y.W.; supervision, G.L. and C.L.; project administration, G.L. All authors have read and agreed to the published version of the manuscript.

**Funding:** This research was funded by National Natural Science Foundation of China (No. 31971400) and the first class General Financial Grant from the China Postdoctoral Science Foundation (No. 2017M620023). We wish to thank the National Key Research and Development Program of China (No. 2017YFC0405303), the Free Flying Wings Program, and the SEE Foundation.

**Acknowledgments:** We would like to thank James Harris and other staff from the International Crane Foundation, Wildlife Science and Conservation Center of Mongolia for the use of the tracking data. We likewise thank Hiroyoshi Higuchi for sharing the early PPT data.

**Conflicts of Interest:** The authors declare no conflict of interest.

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
