# Peer review of "Shifting of the Migration Route of White-Naped Crane (Antigone vipio) Due to Wetland Loss in China"

_remotesensing, doi:10.3390/rs13152984_

Round 1
Reviewer 1 Report
See attached file

Author Response
1:Figure 1. One needs a microscope to understand what’s going on. Yes, I know when I write
papers with my Chinese colleagues that we have to use the official map. But, in this case, we do not need a map of all China. A map of 30 to 50N and 110 to 130 E would work. That would allow the reader to see the all important details.
Response:
I have modified the boundaries of Figure 1 as suggested.
2:Figure 2 is the only other map. It doesn’t capture the essential data that are summarised in Tables 4 and 5. Now, perhaps this doesn’t matter — for what the tables show is the massive increase in urban areas from 1990 to 2010 and the loss of wetlands and grasslands. I think it does matter — we need to see where those losses are. We need a map of wetlands and grasslands and where they were in two time periods. This is surely the core of the paper and presenting that information would be such an improvement.:
Response:
We believe land-use change will provide more information than “habitat” change, including the facts that increased urban land use will increase human disturbance and also lead to greater fragmentation of the overall area. Combining both figures and tables could give more comprehensive information. For instance, combining the changes in the figure and the information in Tables 4 and 5 gave more information than just grasslands and wetlands.
3:A minor point. Using “WNC” for white-naked crane makes reading difficult. In line 70, the could easily write “hereafter just the crane, for simplicity.
Response:
The suggested changes have been done.
Reviewer 2 Report
The paper “Loss of wetlands along the migration routes of White-naped 2 Cranes (Antigone vipio) in China” investigate the relationship among the loss of wetlands comparing two time periods along the migration routes of some birds followed by satellite tracking.
The paper results sound interesting and the research activities and modelling seem properly conducted.
This reviewer suggest to accept the paper for publication on remote Sensing.
Author Response
Thank you for your comments.
Reviewer 3 Report
Review of Remote Sensing MS# 1283988
This study compares the migratory routes and stopover sites of a declining population of the White-naped crane between two year-ranges separated ~20 years apart. It finds that there was a substantial shift of the migration route and stopover areas used by the cranes during this period. The study relates these changes to large-scale human development that converted suitable habitats, especially wetlands, to unsuitable habitats for the cranes. The study is important for understanding flexibility in migration behavior as well as more practically to conserve is rapidly and sharply declining population of this magnificent bird.
Major concerns
- The study aims to compare the habitat utilization of the cranes between two points in time, since in the time interval between these two measurements there were substantial changes in the land cover of the cranes’ stopover area. Yet, from the map (Figure 1) it seems that actually these two periods also correspond to two different sub-populations that breed in different areas. Therefore, how can we be sure that the differences described in the manuscript between the two year-ranges are not just specific patterns characterizing each sub-population? The census data is supporting a shift but this dataset is very limited.
- Since you have the 1990s location data, you need to actually explore the locations that the birds were using in that period since then. Please quantify how these specific localities have changed between 1990s and 2010s and make this analysis explicit for the stopover locations of the birds and not generally for the entire region as a whole. This is how you may provide support for your suggestions that the habitats that were used by the cranes have been altered due to human development.
- There are data from 6 birds in the 1990s and 7 birds from the 2010s. Stopover area increased by ~15% which is in line with the increase in the number of birds tracked in the 2010s since more birds à more stopover area that is used or recorded / higher variability in stopover behavior as more individuals are included in the data. Therefore this increase is expected just by chance due to the larger number of tracked bird. Please respond to this suggestion and explain why it is correct or incorrect.
- Figure 1 – please present only the relevant part of China, Mongolia and Russia – all the western and central parts of China are irrelevant. Please zoom in to the routes of the birds. Also please delete the inset that is found in the lower right corner of the figure – it is not relevant.
- The language needs proof-reading by a native English-speaker – I made numerous amendments to the text (see the file of the manuscript with my edits and comments) but these are only partial corrections of the language and it will require additional reviewing of the text to make it publishable.
Minor comments:
- Flatness is defined as the ratio core area/home range area – if it increase it means that there is less explorative behavior/movement (more core versus home range), or am I missing here something?
- I think that the Discussion is very good, covering the major aspects of this work and the implications for conservation.
Please see an annotated manuscript file for additional comments and edits
